# Oral Microbial Changes in Oral Squamous Cell Carcinoma: Focus on *Treponema denticola*, *Lactobacillus casei*, and *Candida albicans*

**DOI:** 10.3390/medicina60111753

**Published:** 2024-10-25

**Authors:** Yeon-Hee Lee, Junho Jung, Ji-Youn Hong

**Affiliations:** 1Department of Orofacial Pain and Oral Medicine, Kyung Hee University Dental Hospital, #613 Hoegi-dong, Dongdaemun-gu, Seoul 02447, Republic of Korea; 2Department of Oral and Maxillofacial Surgery, School of Dentistry, Kyung Hee University, Dongdaemun-gu, Seoul 02447, Republic of Korea; ssa204@khu.ac.kr; 3Department of Periodontology, Periodontal-Implant Clinical Research Institute, School of Dentistry, Kyung Hee University, Dongdaemun-gu, Seoul 02447, Republic of Korea; jkama7@naver.com

**Keywords:** microbiome, oral squamous cell carcinoma, bacteria, fungi, *Candida albicans*, oral cancer

## Abstract

*Background and Objectives:* In this study, we aimed to explore the oral bacteria and fungi that can help discern oral squamous cell carcinoma (OSCC) and investigate the correlations between multiple key pathogens. *Materials and Methods:* Twelve participants (8 females and 4 males; mean age, 54.33 ± 20.65 years) were prospectively recruited into three groups: Group 1: healthy control, Group 2: patients with stomatitis, and Group 3: patients with OSCC, with 4 individuals in each group. Unstimulated whole saliva samples from these participants were analyzed using real-time PCR to assess the presence and abundance of 14 major oral bacterial species and *Candida albicans*. *Results:* The analysis revealed significant differences for certain microorganisms, namely, *Treponema denticola* (*T. denticola*), *Lactobacillus casei* (*L. casei*), and *Candida albicans*. *T. denticola* was most abundant in the OSCC group (5,358,692.95 ± 3,540,767.33), compared to the stomatitis (123,355.54 ± 197,490.86) and healthy control (9999.21 ± 11,998.40) groups. *L. casei* was undetectable in the healthy control group but was significantly more abundant in the stomatitis group (1653.94 ± 2981.98) and even higher in the OSCC group (21,336.95 ± 9258.79) (*p* = 0.001). A similar trend was observed for *C. albicans*, with DNA copy numbers rising from the healthy control (464.29 ± 716.76) to the stomatitis (1861.30 ± 1206.15) to the OSCC group (9347.98 ± 5128.54) (*p* = 0.006). The amount of *T. denticola* was positively correlated with *L. casei* (r = 0.890, *p* < 0.001) and *C. albicans* (r = 0.724, *p* = 0.008). *L. casei*’s DNA copy number was strongly correlated with *C. albicans* (r = 0.931, *p* < 0.001). These three oral microbes exhibited strong positive correlations with each other and had various direct or indirect relationships with other species. *Conclusions:* In the OSCC group, *T. denticola*, *L. casei*, and *C. albicans* exhibited strong positive correlations with one another, further emphasizing the need for a deeper understanding of the complex microbial interactions in the OSCC environment.

## 1. Introduction

Approximately 38 trillion microorganisms, including bacteria, fungi, viruses, and protozoa, coexist in humans, and their numbers roughly equal those of human cells [1]. These microorganisms are involved in a range of physiological and pathological processes, including cancer development and progression [2]. The concept of intra-oral, salivary, and/or intra-tumoral microbes has emerged to describe microorganisms inhabiting the tumor microenvironment [3,4]. These microbes are located within or adjacent to tumor tissues and have been shown to influence various aspects of tumor biology. Their presence can affect tumor development, progression, and therapeutic responses. The roles of these intra-oral microbes in cancer are gaining recognition as they can modulate the tumor’s biological behavior and interact with the host’s immune system [5]. Understanding these interactions is critical for advancing cancer research and for developing novel therapeutics.

Oral squamous cell carcinoma (OSCC) affects the oral cavity and oropharynx and is the most common form of head and neck cancer. OSCC accounts for more than 90% of cancer cases in this region [6]. It can occur anywhere in the mouth, including the upper and lower gums, palate, floor of the mouth, buccal mucosa, and tongue [7]. Globally, OSCC is one of the most prevalent human malignant tumors, responsible for 1–4% of all cancers and contributing to 2.4% of all cancer-related deaths, reflecting its high mortality rate [8]. Nonetheless, survival rates for oral cancer have significantly improved, increasing by approximately 27% from the mid-1970s to 2018, according to data from the National Institutes of Health [9]. Currently, the overall 5-year survival rate for individuals with oral cancer is 68%, although this rate varies depending on factors such as sex, race, and cancer stage [9]. While smoking and alcohol consumption are well-established risk factors, other causes, such as genetic predisposition and host–microorganism interactions, are still not fully understood. Mounting evidence suggests that oral microbes play crucial roles in the initiation and progression of oral cancer [10,11,12]. Improvements in mortality and treatment outcomes have been supported by advancements in diagnostic tools that allow earlier detection and prevention of disease progression.

Numerous oral microbes have been implicated in OSCC pathogenesis. The oral cavity hosts a diverse and complex microbiome, with more than 700 bacterial species, as well as fungi, viruses, and protozoa, making it second only to the gut in microbial richness [13]. Although the presence of bacteria in human tumors was first documented over a century ago, characterization of the tumor-associated microbiome has proven challenging owing to its low biomass. In 2020, Nejman et al. performed an extensive analysis of bacterial communities in multiple human tumors, such as breast, lung, ovary, pancreas, melanoma, bone, and brain, revealing that each cancer type has a unique bacterial profile [14]. Additionally, the detection of fungi within multiple tumor types highlights the need to further investigate the role of intra-tumoral fungi in cancer diagnosis and prognosis [15]. However, research has not focused on identifying OSCC-specific oral microbial profiles or elucidating the microbial shifts that occur as a healthy oral cavity progresses to a premalignant state or early-stage OSCC. Such investigations are pivotal for advancing our understanding of OSCC, particularly in the areas of early detection and prevention of disease and the development of targeted therapeutic strategies.

Several key factors influence the development and progression of OSCC, including oral and systemic health status, immune responses, and microbial dysbiosis of the oral cavity. This study specifically focused on oral microbes. The primary objective of this study was to present the microbial profiles of patients with OSCC by investigating 15 oral microbes, including 14 predominantly detected oral bacteria and 1 fungus, and to elucidate the correlations among these microbes. The hypothesis of this study was that, instead of a single species being OSCC-specific, certain species may serve as keystone members in forming a pathological microbial network. In addition, we reviewed the significance of OSCC-specific intra-tumoral microbes.

## 2. Materials and Methods

### 2.1. Study Population

Twelve participants (8 females, 4 males; mean age 54.33 ± 20.65 years) were voluntarily recruited for this study at Kyung Hee University Dental Hospital through advertisements between 1 October 2023 and 30 June 2024. The study protocol was approved by the Institutional Review Board of Kyung Hee University Dental Hospital, Seoul, South Korea, in accordance with the Declaration of Helsinki (KHD IRB, IRB No. KH-DT20030, approved on 26 November 2020). Informed consent was obtained from all participants. The participants were divided into three groups: Group 1—healthy controls (3 females and 1 male, 28.25 ± 3.86 years), Group 2—patients with stomatitis (2 females and 2 males, 59.75 ± 5.74 years), and Group 3—patients with OSCC (3 females and 1 male, 75.00 ± 1.82 years). The health status of all participants was assessed by examining oral tissues, including the periodontal tissues and buccal mucosa, as well as general conditions such as oral hygiene and dental calculus deposition.

(1)Inclusion criteria: Participants were required to voluntarily read, understand, and sign the consent form and be capable of participating in the study. Group 1 consisted of medically healthy adults with healthy periodontal and oral mucosal conditions, fewer than two missing teeth in the permanent dentition, and intact oral mucosal integrity. Group 2 comprised patients with stomatitis, characterized by inflammation of the oral mucosa affecting the mouth and lips, with or without oral ulceration. Group 3 comprised patients with OSCC, with inclusion limited to those whose OSCC was confirmed by pathological examination following an incisional biopsy.(2)Exclusion criteria: Individuals with severe xerostomia who could not produce 2 mL of saliva, pregnant or lactating women, adults who failed to comply with clinical examination or sample collection protocols, and those with incomplete data or who withdrew from the study were excluded.

### 2.2. Collection of Unstimulated Whole Saliva

For microbial analysis, 2 mL of unstimulated whole saliva was obtained from all participants using the spitting method. In patients with stomatitis and OSCC, samples were taken prior to any treatment. Participants were asked to avoid caffeine or nicotine for at least 4 h, and alcohol for at least 24 h, before the sampling session. Additionally, they were instructed not to eat, drink, or brush their teeth before saliva collection. Saliva samples were gathered between 9:30 and 11:30 a.m. to reduce diurnal variation, with an average of 3 h between waking and sample collection.

### 2.3. Contamination Prevention

Strict protocols were implemented to minimize contamination during saliva collection and microbial identification. Researchers wore masks, sanitized their hands, and used disinfected gloves, which were replaced between each participant. The experimental workspace was cleaned with alcohol before each session. All equipment and consumables that came into contact with the samples, such as pipette tips and tubes, were sterilized and discarded after a single use. To prevent aerosol contamination, a closed-lid centrifuge was used, and PCR reagent and reaction preparations were conducted on a clean bench [16].

### 2.4. Identification and Quantification of Oral Bacterial and Fungal Species

The quantity, community composition, and individual abundance of bacterial DNA were analyzed. In the saliva samples, the absolute levels and abundance of 14 bacterial species were measured, including *Aggregatibacter actinomycetemcomitans*, *Prevotella intermedia*, *Prevotella nigrescens*, *Eikenella corrodens*, *Campylobacter rectus*, *Fusobacterium nucleatum*, *Porphyromonas gingivalis*, *Treponema denticola*, *Tannerella forsythia*, *Lactobacillus casei*, *Streptococcus mutans*, *Streptococcus sobrinus*, *Parvimonas micra*, and *Eubacterium nodatum*. *Candida albicans* was also identified as a key fungal species [17,18,19]. The specific procedures for identifying and quantifying these bacterial and fungal species are detailed below.

#### 2.4.1. Oral Microbe DNA Isolation

The saliva samples underwent vigorous vortexing to ensure thorough mixing. Then, 500 μL of each sample was combined with 500 μL of lysis buffer (5 mM EDTA, 5 M guanidine hydrochloride, and 0.3 M sodium acetate) in a tube, vortexed, and incubated at 65 °C for 10 min. Following this, 20 μL of S2 buffer (0.25 g/mL silicon dioxide; Merck KGaA, Darmstadt, Germany) was added, and the mixture was vortexed again and incubated at room temperature for 5 min with occasional inversion using an automated system. After centrifugation at 5000 rpm for 30 s, the supernatant was carefully discarded. Next, 1 mL of PureLink (Invitrogen Corporation, Carlsbad, CA, USA) and PCR purification washing buffer 1 (50 mM 3-(N-morpholino) propanesulfonic acid buffer, pH 7.0, with 1 M sodium chloride), activated with 160 mL of 100% ethanol, were added. The solution was vortexed until the beads were fully resuspended. After centrifuging again at 5000 rpm for 30 s, the supernatant was discarded. Then, 1000 μL of ethanol wash buffer 2 was added, followed by vortexing to resuspend the beads. After a final 30 s centrifugation at 5000 rpm, the supernatant was removed. To elute the DNA, 100 μL of elution buffer (100 mM Tris-HCl, pH 7.5, and 1 M EDTA) was added, vortexed, and incubated at 65 °C for 10 min. The isolated DNA was centrifuged at 13,000 rpm for 5 min, and the supernatant was transferred to a sterile microcentrifuge tube for PCR analysis [17,18,19].

#### 2.4.2. Setting the Standard Curve

For setting the standard curve in quantitative real-time PCR (qPCR), it is essential to accurately quantify the DNA copy numbers of target organisms. In this study, a standard curve was generated using plasmid DNA containing the target sequences for each bacterial and fungal species, following previously established methods [17,19,20]. The plasmid DNA concentration was measured using a spectrophotometer, and 10-fold serial dilutions of the plasmid DNA, ranging from 10^1^ to 10^8^ copies per reaction, were prepared to cover a wide range of DNA concentrations [21]. These dilutions were then subjected to qPCR under the same conditions as those applied to the saliva samples. The cycle threshold (Ct) values for each dilution were plotted against the logarithm of the known copy numbers. A linear regression was performed to create the standard curve, which was then used to calculate the DNA copy numbers in the saliva samples [22]. The efficiency of the qPCR reaction was determined from the slope of the standard curve, with only reactions achieving efficiency within the range of 90–110% considered valid for analysis.

#### 2.4.3. Real-Time PCR (qPCR) Amplification

qPCR was carried out to identify and quantify 14 salivary bacterial species using species-specific primers. The total bacterial load was determined with 16S ribosomal RNA (rRNA) primers specific to each bacterium, following the methodology established in our previous work [18]. To measure total bacterial DNA, a universal 16S rRNA primer probe was employed. For *C. albicans*, primers were derived from an earlier study [19] and validated through in silico analysis. A reaction mixture was prepared by combining 5 μL of DNA template, 2.5 μM of each primer pair, and 10 μL of 2X master mix (GeNet Bio, Daejeon, Republic of Korea), with a final volume of 20 μL used for the qPCR. The thermal cycling program involved an initial denaturation at 95 °C for 10 min, followed by 45 cycles of 95 °C for 15 s and 60 °C for 1 min for annealing and extension. Positive controls included plasmid DNA for each target bacterium and *C. albicans*, while DNase/RNase-free water served as the negative control.

#### 2.4.4. Calculation of DNA Copy Numbers for Oral Bacteria and *C. albicans*

To prepare samples, 2 mL of saliva was mixed with 2 mL of a preservation solution. For DNA extraction, 500 μL of this preservation mixture was used, and the final DNA was eluted in 100 μL. qPCR was then performed using 5 μL of the total 100 μL extract. Quantification of DNA was achieved through the standard curve method, utilizing a 5 μL portion for the analysis. DNA copy numbers were calculated for 1 mL of saliva, using the 20 mL preservation solution. The composition of the preservation solution included Tris-HCl, urea, sodium acetate, sodium dodecyl sulfate, EDTA, sodium ascorbate, and ethanol.

#### 2.4.5. α-Diversity Using Shannon’s Diversity Index

Microbial community α-diversity was assessed with Shannon’s diversity index, with bacterial richness quantified by the total number of bacterial DNA copies. The index was calculated using the formula:$$H=-\sum \left(pi\times \ln\left(pi\right)\right)$$

In this equation, *pi* denotes the relative abundance of the *i*th species, determined by n/N, where n is the number of individuals of a given species and N is the total number of individuals in the entire community. Shannon’s index measures diversity, with a score of zero indicating no diversity and higher values reflecting increased biodiversity [17,18].

#### 2.4.6. Statistical Analyses

Descriptive data were expressed as means  ±  standard deviations or as counts and percentages, depending on the type of variable. To assess the distribution of categorical data, χ^2^ tests for proportion equality, Fisher’s exact tests, and Bonferroni corrections were employed. For comparing mean values across groups related to the oral microbiome, analysis of variance (ANOVA) was applied. Spearman’s correlation analysis was used to evaluate relationships between variables, with values closer to ±1 indicating stronger associations. All analyses were conducted using IBM SPSS Statistics (version 26.0; IBM Corp., Armonk, NY, USA) and R (version 4.0.2; R Foundation for Statistical Computing, Vienna, Austria). A two-tailed *p*-value <  0.05 was considered statistically significant.

## 3. Results

### 3.1. Demographics and Oral Examination

Among the three groups in this study, the ages of patients with stomatitis (59.75 ± 5.74 years) and OSCC (75.00 ± 1.82 years) were significantly higher than those of the healthy control group (28.25 ± 3.86 years). Furthermore, the age of the patients with OSCC was significantly higher than that of the patients with stomatitis (*p* < 0.001). There was no statistically significant difference in the female-to-male ratio among the groups, which was 1:1 in the stomatitis group and 3:1 in the OSCC group. Considering the lesion site, visual inspection, X-ray, and CT scans revealed that 100% of the lesions in the stomatitis group were located in the buccal mucosa, while 100% of the lesions in the OSCC group were found in the mandible. During oral examinations, the incidence of periodontitis was significantly higher in the OSCC group than in the healthy control group (0.0% vs. 70%, *p* = 0.018). However, there were no differences between the groups regarding the presence of calculus deposition or poor oral hygiene. In the OSCC group, calculus deposition was observed in 25% of the patients, and poor oral hygiene was noted in 50% of the patients (Table 1).

### 3.2. DNA Copy Numbers of Bacterial and Fungal Species

The DNA copy numbers of 14 bacteria and 1 fungus were compared across the healthy control, stomatitis, and OSCC groups (Table 2). Among these, *T. denticola*, *L. casei*, and *C. albicans* exhibited statistically significant differences between groups (all *p* < 0.05). Specifically, *T. denticola* was found in the highest quantity in the OSCC group (5,358,692.95 ± 3,540,767.33) compared to that in the stomatitis (123,355.54 ± 197,490.86) and healthy control (9999.21 ± 11,998.40) groups, with a significant increase in DNA copy numbers in the stomatitis group relative to the healthy control group (*p* = 0.007). *L. casei* was not detected in the healthy control group, but it was significantly more abundant in the stomatitis group (1653.94 ± 2981.98) and even higher in the OSCC group (21,336.95 ± 9258.79) (*p* = 0.001). A similar pattern was observed for *C. albicans*, with DNA copy numbers increasing from the healthy control (464.29 ± 716.76) to the stomatitis (1861.30 ± 1206.15) to the OSCC group (9347.98 ± 5128.54) (*p* = 0.006). *A. actinomycetemcomitans* was not detected in any group. Aside from *T. denticola*, *L. casei*, and *C. albicans*, no significant quantitative differences were observed among the other 12 microbial species across the groups (Figure 1).

### 3.3. Total Bacterial Amount and Shannon’s Diversity Index

The total bacterial DNA copies, measured using a universal probe for oral bacteria, followed the pattern of healthy control (246,015,329.80 ± 250,216,570.66) < stomatitis (671,981,670.25 ± 645,482,366.81) < OSCC (40,082,611.47 ± 37,609,107.97), with no statistically significant differences (*p* = 0.406). The mean total DNA copies of Gram-negative microorganisms followed the trend of healthy control (1,959,265.36 ± 1,372,737.86) < stomatitis (16,984,317.46 ± 14,157,382.83) < OSCC (26,708,490.05 ± 22,693,303.84), although no statistically significant differences were observed between the groups (*p* = 0.128). Similarly, the total DNA copy numbers of Gram-positive microorganisms increased from the healthy control (92,020.78 ± 82,510.78) to the stomatitis (1861.30 ± 1206.15) to the OSCC group (9347.98 ± 5128.54), yet these differences were not statistically significant (*p* = 0.119). Shannon’s diversity index also increased from the healthy control (1.17 ± 0.35) to the stomatitis (1.42 ± 0.42) to the OSCC group (1.85 ± 1.26), but the difference was not statistically significant (*p* = 0.502) (Table 2 and Figure 1). In general, Shannon’s diversity index falls between 1.5 and 3.5, with values above 4.5 being rare [17].

### 3.4. Prevalence of Bacterial and Fungal Species

The comparison of the prevalence of 15 oral microbes, including 14 oral bacteria and 1 fungus, across the three study groups revealed several significant differences. First, the proportion of *F. nucleatum* relative to the total DNA copies of the 15 oral microorganisms analyzed was significantly higher in the healthy control group (60.47 ± 21.04%) than in the stomatitis (30.82 ± 23.06%) and OSCC groups (7.75 ± 3.62%) (*p* = 0.008). Second, *P. gingivalis* exhibited a significantly different distribution among the groups, with the highest prevalence in the stomatitis group (43.45 ± 24.93%), followed by the OSCC group (16.39 ± 23.18%) and the healthy control group (0.17 ± 0.17%) (*p* = 0.035) (Table 3).

When ranking the mean prevalence of the analyzed microorganisms within each group from highest to lowest, the following patterns were observed:(1)Healthy control group: *F. nucleatum* > *P. nigrescens* > *E. corrodens* > *P. micra* > *T. forsythia* > *C. rectus* > *T. denticola* > *P. intermedia* > *S. mutans* > *P. gingivalis* > *C. albicans* > *E. nodatum* > *S. sanguinis* ≈ *L. casei* > *A. actinomycetemcomitans*.(2)Stomatitis groups were as follows: *P. gingivalis*, *F. nucleatum*, *P. nigrescens*, *T. forsythia* > *E. corrodens*, *C. rectus*, *T. denticola*, *P. micra*, *P. intermedia*, *E. nodatum*, *S. mutans*, *L. casei*, *S. sanguinis*, and *A. actinomycetemcomitans*.(3)OSCC group: *P. micra* > *P. nigrescens* > *E. corrodens* > *T. denticola* > *F. nucleatum* > *P. gingivalis* > *T. forsythia* > *C. rectus* > *S. mutans* > *E. nodatum* > *L. casei* > *C. albicans* > *S. sanguinis* ≈ *P. intermedia* > *A. actinomycetemcomitans*.

Notably, the most prevalent oral bacterial species varied among the groups. *F. nucleatum* was the most prevalent in the healthy control group (60.47%), *P. gingivalis* in the stomatitis group (43.45%), and *P. micra* in the OSCC group (18.85%) (Figure 2A). When the presence of each microbe was analyzed, *T. denticola*, *P. micra*, *L. casei*, *S. mutans*, and *C. albicans* were found to be more prevalent in the OSCC group than in the other groups (Figure 2B).

### 3.5. Correlations Among Oral Microbes

Spearman’s correlation analysis was conducted to examine the relationship between the DNA copies of oral microbes. Importantly, several significant positive correlations were observed between Gram-negative and Gram-positive bacteria and fungi (Table 4).

Based on DNA copy numbers, *T. denticola*, *L. casei*, and *C. albicans* were the oral microbes that exhibited significantly higher levels in the OSCC group; therefore, we focused on the relationships between these species. First, *T. denticola*, *L. casei*, and *C. albicans* showed strong positive correlations. Considering the strength of the correlations with *T. denticola*, *T. denticola* was positively correlated with *L. casei* (r = 0.890, *p* < 0.001), *E. corrodens* (r = 0.752, *p* = 0.005), *P. micra* (r = 0.742, *p* = 0.006), *C. albicans* (r = 0.724, *p* = 0.008), *P. nigrescens* (r = 0.708, *p* = 0.010), and *T. forsythia* (r = 0.634, *p* = 0.027). Species that were positively correlated with *L. casei* DNA copy numbers presented with the following correlation strengths: *C. albicans* (r = 0.931, *p* < 0.001) > *T. denticola* (r = 0.890, *p* < 0.001) > *P. nigrescens* (r = 0.849, *p* < 0.001) > *P. micra* (r = 0.889, *p* < 0.001) > *E. corrodens* (r = 0.849, *p* < 0.001), *E. nodatum* (r = 0.751, *p* = 0.005) > *C. rectus* (r = 0.653, *p* = 0.021). The oral microbes that were positively correlated with *C. albicans* DNA copy numbers were *L. casei* (r = 0.931, *p* < 0.001), *P. micra* (r = 0.913, *p* < 0.001), *E. corrodens* (r = 0.909, *p* < 0.001), *P. nigrescens* (r = 0.878, *p* < 0.001), *C. rectus* (r = 0.790, *p* = 0.002), *T. denticola* (r = 0.724, *p* = 0.008), and *E. nodatum* (r = 0.674, *p* = 0.016). Additionally, *P. micra*, which was the most abundant in the OSCC group compared to that in the healthy control and stomatitis groups, was positively correlated with *E. corrodens* (r = 0.998, *p* < 0.001), *P. nigrescens* (r = 0.974, *p* < 0.001), *C. albicans* (r = 0.931, *p* < 0.001), *L. casei* (r = 0.889, *p* < 0.001), *C. rectus* (r = 0.761, *p* = 0.004), and *T. denticola* (r = 0.742, *p* = 0.006). Notably, no significant negative correlations were observed among the microbes, suggesting an inhibition or reduction in the presence of certain species.

Deeper shades of red indicate correlations closer to +1, while deeper shades of blue signify correlations closer to −1. *Aggregatibacter actinomycetemcomitans* was not detected in the real-time PCR and was therefore excluded from this correlation analysis.

### 3.6. In-Depth Analysis of T. denticola, C. albicans, and L. casei

An in-depth analysis based on the schematic diagram illustrating the direct and indirect relationships between the three prominent microbes in the OSCC group—*T. denticola*, *C. albicans*, and *L. casei*—revealed the following findings: *T. denticola* was directly correlated with Gram-negative species such as *P. nigrescens*, *E. corrodens*, and *P. gingivalis*. *F. nucleatum* exhibited an indirect positive correlation with *T. denticola* via *P. nigrescens*. Although *C. rectus* was not directly correlated with *T. denticola*, it was indirectly correlated with *P. nigrescens*, *E. corrodens*, *T. forsythia*, and *P. gingivalis*. Among the Gram-positive species, *T. denticola* showed a direct positive correlation with *L. casei* and *P. micra*. A strong, direct positive correlation was observed between *T. denticola* and *C. albicans* (Figure 3A). *C. albicans* was positively correlated with several Gram-negative species, including *T. denticola*, *E. corrodens*, *P. nigrescens*, and *C. rectus*. Among the Gram-positive species, *C. albicans* was significantly and positively correlated with *P. micra*, *E. nodatum*, and *L. casei* (Figure 3B). *L. casei* was positively correlated with several Gram-positive species, including *P. nigrescens*, *E. corrodens*, *P. micra*, and *E. nodatum*. Among the Gram-negative species, *L. casei* had significant positive correlations with *T. denticola* and *C. rectus* also showed a strong positive correlation with *C. albicans* (Figure 3C).

## 4. Discussion

In this study, we investigated the distribution of 14 major oral bacteria and *C. albicans* across three groups: healthy controls, patients with stomatitis, and patients with OSCC. Our findings revealed that *T. denticola*, *L. casei*, and *C. albicans* were significantly more abundant in the OSCC group than in the other groups (Figure 4). Specifically, *T. denticola* was present in the highest quantity in the OSCC group, followed by the stomatitis and healthy control groups. *L. casei* was undetectable in healthy controls but showed a marked increase in both the stomatitis and OSCC groups. Similarly, *C. albicans* exhibited a progressive increase in DNA copy numbers from healthy controls to patients with stomatitis and those with OSCC. Considering the correlations among oral microbes, a wide range of positive associations were observed between Gram-negative and Gram-negative bacteria, Gram-positive and Gram-positive bacteria, Gram-negative and Gram-positive bacteria, *C. albicans* and Gram-negative bacteria, and *C. albicans* and Gram-positive bacteria. OSCC-associated microbes engage in quorum sensing, a key mechanism influencing the development and progression of oral cancers. However, microbial diversity, as indicated by Shannon’s diversity index, showed no significant differences across the groups. These findings point to a potential link between these specific microbes and OSCC, highlighting the need for further exploration of their role as potential OSCC biomarkers.

Among the Gram-negative species, *T. denticola* was notably more abundant in the OSCC group than in the healthy control and stomatitis groups. *T. denticola* is a motile, obligate anaerobic bacterium and highly proteolytic spirochete that lives in the oral cavity of humans and is predominantly found in periodontal lesions associated with adult periodontitis [23]. *T. denticola*, along with *P. gingivalis* and *F. nucleatum*, forms a red complex that plays a crucial role in periodontal disease. *P. gingivalis* has been recognized as a significant pathogen not only in periodontal disease but also in the development of OSCC [24,25]. In this study, although the DNA copy number of *P. gingivalis* was higher in patients with stomatitis (7,803,707.73 ± 10,628,172.40) and OSCC (2,501,079.28 ± 1,786,087.94) compared to healthy controls (2076.07 ± 1336.47), the difference was not statistically significant. However, with a larger sample size in future studies, we expect that this difference may become statistically significant. Numerous studies have linked periodontitis to OSCC, and periodontal pathogens such as *T. denticola* have been implicated in OSCC pathogenesis [11,26]. Additionally, *T. denticola* has been shown to directly promote OSCC cell proliferation through activation of the intracellular TGF-β pathway [11]. In this study, *T. denticola* showed direct correlations with Gram-negative species like *P. gingivalis*, *P. nigrescens*, and *E. corrodens*. Notably, *P. nigrescens* serves as a connector between *T. denticola* and *F. nucleatum*. Recent research has found an increased abundance of Fusobacteria at the phylum level, *Fusobacterium* at the genus level, and *F. nucleatum* at the species level in OSCC patients [27]. *F. nucleatum*, a key pro-tumorigenic bacterium, tends to accumulate at the invasive edges of OSCC tissues, promoting tumor-associated macrophage formation [28]. However, the exact involvement of these bacteria in OSCC remains unclear, highlighting the need for further studies to better understand their roles in the disease’s prognosis and pathogenesis.

*T. denticola* exhibited a direct positive correlation with *L. casei* and *P. micra* and a strong direct positive correlation with *C. albicans*. Additionally, several direct and indirect complex interactions among the oral microbes were observed. Avril et al. proposed a “bacterial driver–passenger” model in colorectal cancer, wherein initial “driver” bacteria induce alterations in the microenvironment, thereby facilitating tumor formation [29]. In the early stages of oral cancer, key pathogens increase, followed by subsequent changes in the associated microbial communities [30]. As the tumor advances, these driver bacteria are progressively replaced by “passenger” bacteria, which include opportunistic pathogens and commensal or probiotic species. These passenger bacteria can further impact tumor development by interacting with both the host and the tumor microenvironment [31]. In patients with OSCC, additional research is required to elucidate driver–passenger microbe relationships and delineate the sequential dynamics of microbial populations within the tumor context.

Among the Gram-positive bacteria, *L. casei* has emerged as the most prominent species associated with OSCC. *Lactobacilli* are Gram-positive, anaerobic, rod-shaped commensal bacteria typically found in the oral, genitourinary, and gastrointestinal tracts of humans [32]. *Lactobacillus* is among the most frequently detected bacterial genera in the saliva of OSCC patients. [33,34]. *L. casei* was positively correlated with several Gram-positive species, including *P. nigrescens*, *E. corrodens*, *P. micra*, and *E. nodatum*. Among the Gram-negative species, *L. casei* had a significantly strong positive correlation with *T. denticola* and *C. rectus*. Notably, members of the *Lactobacillus* genus exhibit both carcinogenic and anticarcinogenic properties. The secretion of lactic acid and other organic acids by these bacteria can lower the pH of the tumor microenvironment, potentially facilitating OSCC progression [35]. Moreover, changes in the activities of nicotinamide adenine dinucleotide phosphate oxidase and nitric oxide synthase can result in the buildup of reactive oxygen and nitrogen species, contributing to sustained inflammation [36]. Some *Lactobacillus* species may exacerbate this process by generating hydrogen peroxide [37]. However, it is crucial to recognize that, while *Lactobacillus* species are prevalent in individuals with OSCC, their presence does not necessarily imply a causative role in the disease [38]. Further research is required to fully elucidate the roles of *Lactobacillus* and other microbes in OSCC development.

*C. albicans* positively correlated with several Gram-negative species, including *T. denticola*, *E. corrodens*, *P. nigrescens*, and *C. rectus*. This study focused on *C. albicans*, which is the fungus most commonly associated with oral diseases. *C. albicans* can cause oral candidiasis, commonly known as thrush [39]. While the bacterial elements of the OSCC microbiome have been extensively studied, the fungal components remain less explored and poorly understood [40]. Nonetheless, *Candida*, a genus of yeast-like fungi, continues to be a prominent subject of study in OSCC. Though *C. albicans* is the dominant species, other *Candida* species have also been identified in OSCC patients [33,41]. Research indicates that 72.2% of these patients harbor *Candida* species in their saliva [33]. The carcinogenic mechanisms of *C. albicans* are intricate, with the initial phase involving its attachment to mucosal epithelial cells, which serve as the body’s first line of defense against pathogens [42]. This adhesion can compromise the host’s immune defenses, facilitating further infection. Additionally, *C. albicans* produces carcinogens, such as nitrosamines, which can activate proto-oncogenes and induce carcinomatous changes [43]. Chronic inflammation associated with *C. albicans* infection may also contribute to cancer development [44]. Furthermore, *C. albicans* may promote carcinogenesis by stimulating the T helper 17 (Th17) response. Th17 cells play a crucial role in maintaining mucosal barriers and eliminating pathogens from mucosal surfaces [45]. *C. albicans* exhibited significant positive correlations with *L. casei*, *P. micra*, and *E. nodatum* among Gram-positive species. However, these complex interactions remain underexplored, and further research employing advanced statistical and analytical methods is required to fully elucidate these relationships.

In this study, although the mean Shannon’s index, which is an index of α-diversity, increased from healthy controls to stomatitis patients and then to OSCC patients, the differences were not statistically significant. Consistent with the results of Zhao et al. [46], oral cancer samples exhibited a greater variety of bacterial species than did healthy control samples. This finding is supported by recent reviews that have reported higher α-diversity in OSCC patient samples than in healthy controls [27]. Additionally, patients with OSCC and a history of tobacco use showed an increase in α-diversity [47]. Studies have similarly indicated that cancers at various sites often present with elevated α-diversity in the microbiome compared with that in healthy controls [48]. This increase in microbial diversity may be attributed to several factors, including a chronic inflammatory environment, alterations in local pH and nutrient availability, changes in tumor-associated inflammatory responses, and environmental conditions. In the current study, no specific microbial species were found to be directly correlated with the increase in Shannon’s diversity index observed in OSCC. Future research with a larger cohort of OSCC samples is needed to identify the microbes associated with microbial diversity and to further investigate changes in microbial profiles relative to healthy controls.

A limitation of this preliminary study is its small sample size of only 12 participants, including only 4 patients with OSCC. As the age range of healthy controls was significantly younger than that of other oral disease groups, which may have led to misinterpretation or bias in our findings, future studies with age-matched healthy controls will be necessary to strengthen the validity of our results. Additionally, we did not utilize advanced analytical techniques such as next-generation sequencing or state-of-the-art AI-driven analyses. Although *L. casei* is commonly found in fermented milk products and is often included in the composition of probiotics [49,50], we did not exclude individuals who consumed fermented milk products or *L. casei*-containing probiotics from the study. Despite these limitations, this study has several strengths. We concurrently assessed both the major bacterial species and fungi using qPCR, which allowed for a comprehensive analysis of the oral microbial profile and the interrelationships within the same saliva samples. This methodology enabled us to investigate the roles of specific microbes and their complex interactions in the pathological mechanisms of OSCC. Previous studies targeting OSCC-specific pathogens have often concentrated on individual microbes, such as *P. gingivalis* or *T. denticola* [10,11,51,52]. Although our preliminary results are encouraging, further studies with larger sample sizes and multicenter designs are essential to validate these findings. Follow-up studies are being planned to address these issues.

## Figures and Tables

**Figure 1 medicina-60-01753-f001:**
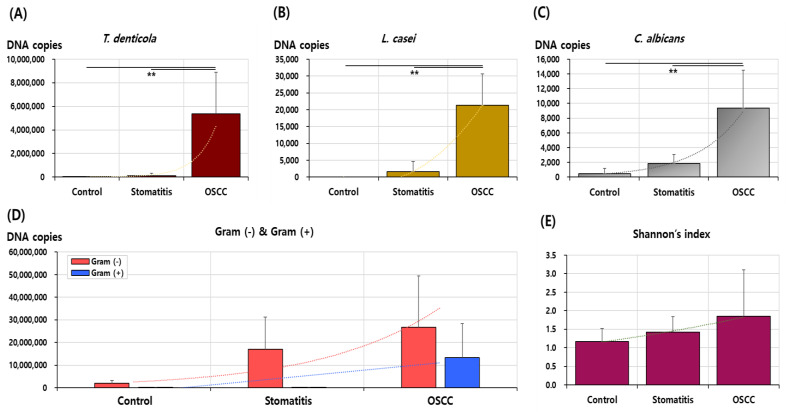
Comparison of DNA copy numbers of oral bacteria and *Candida albicans* between groups. The DNA copy numbers of (**A**) *T. denticola*, (**B**) *L. casei*, (**C**) *C. albicans*, and (**D**) Gram-negative and Gram-positive species were evaluated. (**E**) Shannon’s index was analyzed. All statistical analyses were conducted using analysis of variance, with statistical significance defined at *p* < 0.05. Results marked with ** indicate a significance level of *p* < 0.01.

**Figure 2 medicina-60-01753-f002:**
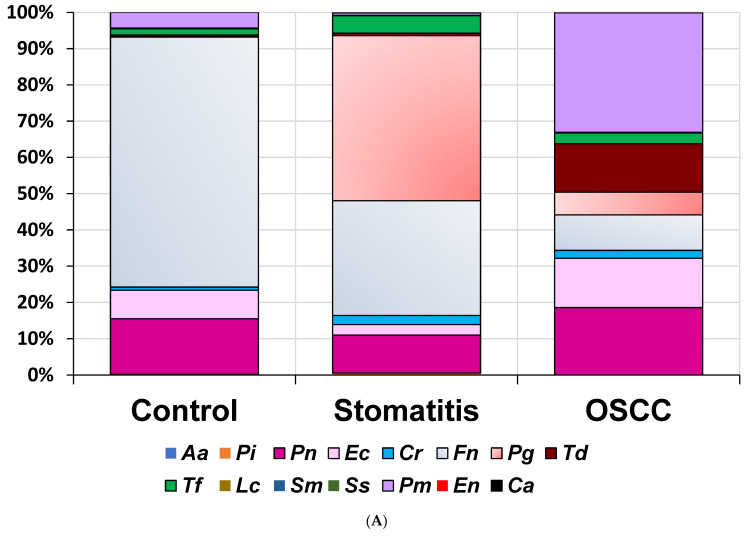
Group-wise comparison of the prevalence of each oral microbe. (**A**) Prevalence of each species by group. (**B**) Percentage distribution of groups for each microbe. *Aa*, *Aggregatibacter actinomycetemcomitans*; *Pi*, *Prevotella intermedia*; *Pn*, *Prevotella nigrescens*; *Ec*, *Eikenella corrodens*; *Cr*, *Campylobacter rectus*; *Fn*, *Fusobacterium nucleatum*; *Pg*, *Porphyromonas gingivalis*; *Td*, *Treponema denticola*; *Tf*, *Tannerella forsythia*; *Lc*, *Lactobacillus casei*; *Sm*, *Streptococcus mutans*; *Ss*, *Streptococcus sobrinus*; *Pm*, *Parvimonas micra*; *En*, *Eubacterium nodatum*; *Ca*, *Candida albicans*.

**Figure 3 medicina-60-01753-f003:**
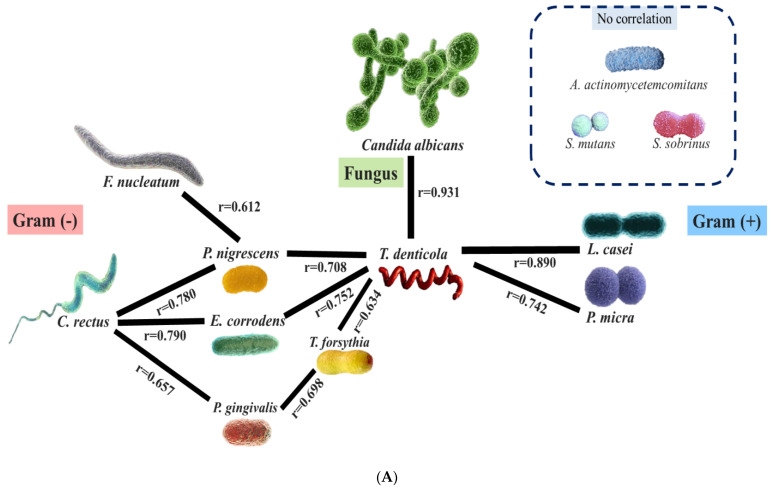
Schematic presentation of the relationships between oral microbes. (**A**) Microbial linkages specific to *T. denticola*. (**B**) Microbial linkages specific to *Candida albicans*. (**C**) Microbial linkages specific to *L. casei*. Based on the results of Spearman’s correlation analysis, the diagram illustrates the microbial relationships. The thickness of each bar reflects the strength of the correlation between two oral microbes. *P. intermedia*: *Prevotella intermedia*; *P. nigrescens*: *Prevotella nigrescens*; *E. corrodens*: *Eikenella corrodens*; *C. rectus*: *Campylobacter rectus*; *F. nucleatum*: *Fusobacterium nucleatum*; *P. gingivalis*: *Porphyromonas gingivalis*; *T. denticola*: *Treponema denticola*; *T. forsythia*: *Tannerella forsythia*; *L. casei*: *Lactobacillus casei*; *S. mutans*: *Streptococcus mutans*; *S. sobrinus*: *Streptococcus sobrinus*; *P. micra*: *Parvimonas micra*; *E. nodatum*: *Eubacterium nodatum*; *C. albicans*: *Candida albicans*.

**Figure 4 medicina-60-01753-f004:**
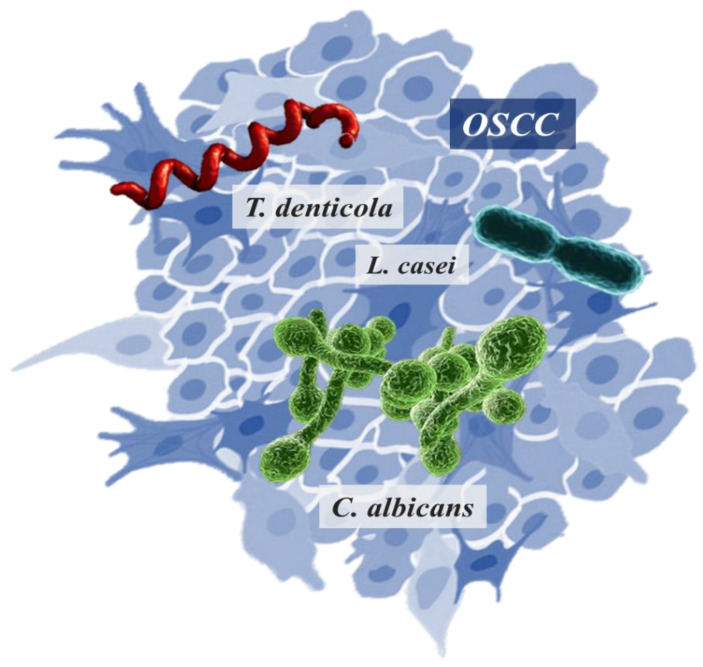
Oral microbes associated with oral squamous cell carcinoma. *T. denticola: Treponema denticola*; *L. casei: Lactobacillus casei*; *C. albicans: Candida albicans*; OSCC: oral squamous cell carcinoma.

**Table 1 medicina-60-01753-t001:** Demographics and oral examination.

Parameters	Healthy Control Mean ± SD or *n* (%)	Stomatitis Mean ± SD or *n* (%)	OSCC Mean ± SD or *n* (%)	*p*-Value	Post Hoc
Age (years) ^a^	28.25 ± 3.86	59.75 ± 5.74	75.00 ± 1.82	**<0.001 *****	Healthy control < Stomatitis and OSCC, Stomatitis < OSCC
Sex ^b^					
Female	3 (75.0%)	2 (50.0%)	3 (75.0%)	0.687	
Male	1 (25.0%)	2 (50.0%)	1 (25.0%)		
Lesion site					
Buccal mucosa ^b^	0 (0.0%)	4 (100.0%)	0 (0.0%)	**<0.001 *****	
Mandible ^b^	0 (0.0%)	0 (0.0%)	4 (100.0%)	**<0.001 *****	
Oral examination					
Periodontitis ^b^	0 (0.0%)	2 (50.0%)	3 (75.0%)	**0.018 ***	
Calculus deposition ^b^	0 (0.0%)	1 (25.0%)	1 (25.0%)	0.549	
Poor oral hygiene ^b^	0 (0.0%)	0 (0.0%)	2 (50.0%)	0.091	

The results were obtained using ^a^ analysis of variance (ANOVA) and post hoc analysis. ^b^ Chi-square test (two-sided). Statistical significance was set at *p* < 0.05. Significant differences are indicated in bold. * *p* < 0.05, *** *p* < 0.001; SD: standard deviation; Healthy control: healthy control participants; Stomatitis: patients with stomatitis; OSCC: patients with oral squamous cell carcinoma.

**Table 2 medicina-60-01753-t002:** Comparison of DNA copies of bacterial and fungal species, and Shannon’s diversity index.

Microbial Parameter		Healthy Control Mean ± SD (Copies)	Stomatitis Mean ± SD (Copies)	OSCC Mean ± SD (Copies)	*p*-Value	Post Hoc
Gram (−)	*Aa*	0 ± 0	0 ± 0	0 ± 0	NA	
	*Pi*	4721.43 ± 9338.45	97,123.55 ± 163,965.30	0 ± 0	0.311	
	*Pn*	313,182.60 ± 162,170.03	1,783,400.78 ± 1,859,234.18	7,444,085.88 ± 7,776,100.68	0.124	
	*Ec*	161,638.36 ± 73,445.12	497,355.67± 376,410.96	5,455,613.30 ± 5,929,438.11	0.101	
	*Cr*	17,702.63 ± 19,738.29	431,907.02 ± 725,172.58	869,577.08 ± 745,262.35	0.189	
	*Fn*	1,414,677.53 ± 1,248,019.45	5,425,576.88 ± 6,134,874.96	3,928,871.25 ± 4,209,210.50	0.453	
	*Pg*	2076.07 ± 1336.47	7,803,707.73 ± 10,628,172.40	2,501,079.28 ± 1,786,087.94	0.247	
	*Td*	9999.21 ± 11,998.40	123,355.54 ± 197,490.86	5,358,692.95 ± 3,540,767.33	**0.007 ****	Healthy control < Stomatitis and OSCC, Stomatitis < OSCC
	*Tf*	35,267.53 ± 25,096.14	821,889.97 ± 910,730.89	1,150,570.3 ± 721,737.42	0.106	
Gram (+)	*Lc*	0 ± 0	1653.94 ± 2981.98	21,336.95 ± 9258.79	**0.001 ****	Healthy control < Stomatitis and OSCC, Stomatitis < OSCC
	*Sm*	3258.15 ± 3492.39	18,238.71 ± 35,986.69	100,785.83 ± 181,256.31	0.416	
	*Ss*	0 ± 0	641.81 ± 1283.62	0 ± 0	0.405	
	*Pm*	88,593.26 ± 81,242.44	112,672.78 ± 85,587.27	13,204,631.90 ± 15,038,585.05	0.098	
	*En*	169.37 ± 226.55	22,360.99 ± 32,330.07	38,018.78 ± 37,252.07	0.223	
Fungus	*Ca*	464.29 ± 716.76	1861.30 ± 1206.15	9347.98 ± 5128.54	**0.006 ****	Healthy control < Stomatitis and OSCC, Stomatitis < OSCC
Gram (−)		1,959,265.36 ± 1,372,737.86	16,984,317.46 ± 14,157,382.83	26,708,490.05 ± 22,693,303.84	0.128	
Gram (+)		92,020.78 ± 82,510.78	155,568.13 ± 131,036.11	13,364,773.45 ± 14,988,291.76	0.093	
DNA copies of 15 microbes		2,051,750.43 ± 1,394,889.68	17,141,746.89 ± 14,070,899.96	40,082,611.47 ± 37,609,107.97	0.119	
Total bacteria		246,015,329.80 ± 250,216,570.66	671,981,670.25 ± 645,482,366.81	827,473,263.93 ± 781,237,168.13	0.406	
Shannon’s diversity index		1.17 ± 0.35	1.42 ± 0.42	1.85 ± 1.26	0.502	

The results were obtained using analysis of variance (ANOVA) and post hoc analysis. Differences between groups were considered significant at *p* < 0.05. Significant differences are indicated in bold. ** *p* < 0.01; SD: standard deviation; Healthy control: healthy control participants; Stomatitis: patients with stomatitis; OSCC: patients with oral squamous cell carcinoma. *Aa*, *Aggregatibacter actinomycetemcomitans*; *Pi*, *Prevotella intermedia*; *Pn*, *Prevotella nigrescens*; *Ec*, *Eikenella corrodens*; *Cr*, *Campylobacter rectus*; *Fn*, *Fusobacterium nucleatum*; *Pg*, *Porphyromonas gingivalis*; *Td*, *Treponema denticola*; *Tf*, *Tannerella forsythia*; *Lc*, *Lactobacillus casei*; *Sm*, *Streptococcus mutans*; *Ss*, *Streptococcus sobrinus*; *Pm*, *Parvimonas micra*; *En*, *Eubacterium nodatum*; *Ca*, *Candida albicans.* Gram (−): Gram-negative species (*Aa*, *Pi*, *Pn*, *Ec*, *Cr*, *Fn*, *Pg*, *Td*, and *Tf*); Gram (+): Gram-positive species (*Lc*, *Sm*, *Ss*, *Pm*, and *En*).

**Table 3 medicina-60-01753-t003:** Comparison of the prevalence rate (%) of 14 oral bacterial and 1 fungal species.

Microbial Parameters		Healthy Control Mean ± SD (%)	Stomatitis Mean ± SD (%)	OSCC Mean ± SD (%)	*p*-Value	Post Hoc
Gram (−)	*Aa*	0 ± 0	0 ± 0	0 ± 0	NA	
	*Pi*	0.47 ± 0.93	1.37 ± 1.86	0 ± 0	0.311	
	*Pn*	19.88 ± 16.99	11.09 ± 6.83	16.91 ± 7.35	0.561	
	*Ec*	9.65 ± 6.59	2.78 ± 1.55	8.64 ± 7.23	0.239	
	*Cr*	1.37 ± 2.07	1.53 ± 1.94	4.17 ± 5.11	0.451	
	*Fn*	60.47 ± 21.04	30.82 ± 23.06	7.75 ± 3.62	**0.008 ****	Healthy control > Stomatitis > OSCC
	*Pg*	0.17 ± 0.17	43.45 ± 24.93	16.39 ± 23.18	**0.035 ***	Healthy control < Stomatitis, OSCC < Stomatitis
	*Td*	1.05 ± 1.28	1.32 ± 2.33	20.96 ± 27.89	0.192	
	*Tf*	1.98 ± 1.72	4.74 ± 2.31	5.28 ± 5.15	0.379	
Gram (+)	*Lc*	0 ± 0	0.05 ± 0.09	0.14 ± 0.15	0.195	
	*Sm*	0.22 ± 0.27	0.58 ± 1.15	0.77± 1.52	0.783	
	*Ss*	0 ± 0	0.02 ± 0.04	0 ± 0	0.405	
	*Pm*	4.68 ± 4.26	1.67 ± 2.04	18.85 ± 20.05	0.148	
	*En*	0.01 ± 0.01	0.57 ± 1.08	0.07 ± 0.05	0.415	
Fungus	*Ca*	0.04 ± 0.07	0.02 ± 0.02	0.07 ± 0.10	0.658	

The results were obtained using analysis of variance (ANOVA) and post hoc analysis. Differences between groups were considered significant at *p* < 0.05. Significant differences are indicated in bold. * *p* < 0.05, ** *p* < 0.01; SD: standard deviation; Healthy control: healthy control participants; Stomatitis: patients with stomatitis; OSCC: patients with oral squamous cell carcinoma. *Aa*, *Aggregatibacter actinomycetemcomitans*; *Pi*, *Prevotella intermedia*; *Pn*, *Prevotella nigrescens*; *Ec*, *Eikenella corrodens*; *Cr*, *Campylobacter rectus*; *Fn*, *Fusobacterium nucleatum*; *Pg*, *Porphyromonas gingivalis*; *Td*, *Treponema denticola*; *Tf*, *Tannerella forsythia*; *Lc*, *Lactobacillus casei*; *Sm*, *Streptococcus mutans*; *Ss*, *Streptococcus sobrinus*; *Pm*, *Parvimonas micra*; *En*, *Eubacterium nodatum*; *Ca*, Candida albicans. Gram (−): Gram-negative species (*Aa*, *Pi*, *Pn*, *Ec*, *Cr*, *Fn*, *Pg*, *Td*, and *Tf*); Gram (+): Gram-positive species (*Lc*, *Sm*, *Ss*, *Pm*, and *En*).

**Table 4 medicina-60-01753-t004:** Correlation between DNA copies of oral microbes.

Correlation Coefficient		Gram (−)							Gram (+)					Fungus
		*Pn*	*Ec*	*Cr*	*Fn*	*Pg*	*Td*	*Tf*	*Lc*	*Sm*	*Ss*	*Pm*	*En*	*Ca*
Gram (−)	*Pi*	−0.193	−0.160	−0.215	−0.129	−0.006	−0.176	−0.075	−0.233	−0.115	0.045	−0.163	−0.151	−0.112
	*Pn*		**0.980**	**0.780**	**0.612**	0.097	**0.708**	0.498	**0.849**	−0.130	−0.165	**0.974**	**0.761**	**0.878**
	*Ec*			**0.790**	0.465	0.087	**0.752**	0.522	**0.884**	−0.091	−0.158	**0.998**	**0.755**	**0.909**
	*Cr*				0.410	**0.657**	0.515	**0.788**	**0.653**	−0.175	−0.208	**0.761**	0.497	**0.790**
	*Fn*					0.183	0.198	0.290	0.243	−0.264	−0.220	0.434	0.372	0.277
	*Pg*						−0.068	**0.698**	−0.058	−0.167	−0.086	0.036	−0.051	0.120
	*Td*							**0.634**	**0.890**	0.575	−0.180	**0.742**	0.572	**0.724**
	*Tf*								0.524	0.337	−0.220	0.472	0.296	0.522
Gram (+)	*Lc*									0.265	−0.043	**0.889**	**0.751**	**0.931**
	*Sm*										0.093	−0.104	0.049	−0.038
	*Ss*											−0.134	0.500	−0.138
	*Pm*												**0.766**	**0.913**
	*En*													**0.674**

The results were obtained through Spearman’s correlation analysis. Differences between groups were considered significant at *p*-value < 0.05. Statistically significant results are highlighted in bold. *Pi*, *Prevotella intermedia*; *Pn*, *Prevotella nigrescens*; *Ec*, *Eikenella corrodens*; *Cr*, *Campylobacter rectus*; *Fn*, *Fusobacterium nucleatum*; *Pg*, *Porphyromonas gingivalis*; *Td*, *Treponema denticola*; *Tf*, *Tannerella forsythia*; *Lc*, *Lactobacillus casei*; *Sm*, *Streptococcus mutans*; *Ss*, *Streptococcus sobrinus*; *Pm*, *Parvimonas micra*; *En*, *Eubacterium nodatum*; *Ca*, *Candida albicans*. Gram (−): Gram-negative species (*Pi*, *Pn*, *Ec*, *Cr*, *Fn*, *Pg*, *Td*, and *Tf*); Gram (+): Gram-positive species (*Lc*, *Sm*, *Ss*, *Pm*, and *En*).

## Data Availability

The datasets used and/or analyzed in the current study are available from the corresponding author upon reasonable request.

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
