# Peer review of "Oral Microbial Changes in Oral Squamous Cell Carcinoma: Focus on *Treponema denticola*, *Lactobacillus casei*, and *Candida albicans"

_medicina, 2024, doi:10.3390/medicina60111753_

Round 1
Reviewer 1 Report
Comments and Suggestions for Authors
The authors compared oral microbe compositions among healthy, stomatitis, and OSCC subjects by collecting the salivary and PCR examinations. They found that the increased Treponema denticola (T. denticola), Lactobacillus casei (L. casei), and Candida albicans in OSCC patients. This study exists several concerns.
-
1. The title of the article is exaggerated and the methodology of the study, which only used saliva for bacteriological analysis, can only be described as 'intra-oral' and should not be described as 'intra-tumour'.
2. L. casei is commonly found in fermented milk products, did the authors exclude people who consumed fermented milk products when recruiting the study subjects? It lacks a clear description.
3. Porphyromonas gingivalis is the key factor in periodontal disease development, especially in OSCC development (BMC Oral Health. 2021. 3;21(1):228.). But it is surprise to see that there was no difference among groups. It seems to be not clear for the method in the calculation of DNA copy numbers (no information about the method for setting standard curve)
4. The age range was too different between healthy and other oral disease groups, which could lead to misinterpretation.
5. How Table 3 values, which was very differ from Tbale2 for the same bacteria or fungus, are calculated and what they represent is not clearly explained. The lack of clear explanation for these discrepancies suggests either carelessness or deliberate obfuscation.
English is fine.
Author Response
Reviewer 1
The authors compared oral microbe compositions among healthy, stomatitis, and OSCC subjects by collecting the salivary and PCR examinations. They found that the increased Treponema denticola (T. denticola), Lactobacillus casei (L. casei), and Candida albicans in OSCC patients. This study exists several concerns.
Response:
We sincerely appreciate that you have recognized and acknowledged the core of our research. In response to your suggestion, the revised sections have been highlighted in red.
- The title of the article is exaggerated and the methodology of the study, which only used saliva for bacteriological analysis, can only be described as 'intra-oral' and should not be described as 'intra-tumour'.
Response:
We agree with your opinion and have revised the title of our study as follows:
Intra-Oral Microbes in Oral Squamous Cell Carcinoma: Focus on Treponema denticola, Lactobacillus casei, and Candida albicans
- L. casei is commonly found in fermented milk products, did the authors exclude people who consumed fermented milk products when recruiting the study subjects? It lacks a clear description.
Response:
Previous human studies that focused on L. casei did not necessarily exclude participants who consumed fermented milk products or probiotics containing L. casei. However, we have taken your valuable suggestion into consideration and have clarified this aspect in Discussion section of our manuscript. Thank you for your insightful feedback.
- Porphyromonas gingivalis is the key factor in periodontal disease development, especially in OSCC development (BMC Oral Health. 2021. 3;21(1):228.). But it is surprise to see that there was no difference among groups. It seems to be not clear for the method in the calculation of DNA copy numbers (no information about the method for setting standard curve).
Response:
We sincerely appreciate your constructive comments and suggestion. P. gingivalis plays a key role in the development of OSCC from periodontal disease and can independently influence the mechanisms underlying OSCC. In our study, while the DNA copy number of P. gingivalis between OSCC patients and healthy controls did not show a statistically significant difference, the mean value was higher in patients with stomatitis (7803707.73 ± 10628172.40) or OSCC (2501079.28 ± 1786087.94) compared to healthy controls (2076.07 ± 1336.47) (p=0.247). Additionally, Candida albicans has recently gained attention as a microorganism associated with OSCC development, and it has been studied independently as well. We anticipate that, with an increased sample size, these differences could potentially reach statistical significance.
We have added the following sentences to the discussion section of the manuscript.
- gingivalis has been identified as a key pathogen not only in periodontal disease but also in the development of OSCC [22,23]. In this study, the DNA copy number of P. gingivalis was higher in patients with stomatitis (7,803,707.73 ± 10,628,172.40) or OSCC (2,501,079.28 ± 1,786,087.94) compared to healthy controls (2,076.07 ± 1,336.47), although this difference was not statistically significant. We anticipate that with an increased sample size in future studies, this difference may reach statistical significance.
We have added the method for setting the standard curve to the Method section as per your suggestion. Thank you for your valuable input.
- The age range was too different between healthy and other oral disease groups, which could lead to misinterpretation.
Response:
Thank you for your comments. We acknowledge that this is a clear limitation of our study. However, we kindly ask for your understanding regarding the practical challenges of enrolling age-matched healthy controls. We recognize that this factor may introduce bias into the data, and we plan to conduct additional studies with age- and sex-matched healthy controls in the future. Thank you for your understanding in advance.
- How Table 3 values, which was very differ from Tbale2 for the same bacteria or fungus, are calculated and what they represent is not clearly explained. The lack of clear explanation for these discrepancies suggests either carelessness or deliberate obfuscation.
Response:
Table 2 presents the statistical results for the DNA copy number of bacteria and fungi, while Table 3 shows the proportion (%) of each bacterium and fungus relative to the total DNA copies of the 15 oral microorganisms. In other words, the numbers in Table 2 represent the absolute DNA copy numbers, whereas the numbers in Table 3 represent the relative composition percentages (%) of the total. To clarify this distinction, we have added the "%" symbol in Table 3. Thank you for your valuable comments.

Reviewer 2 Report
Comments and Suggestions for Authors
The chosen topic is particularly interesting. The study is well completed, in which the methodical work can be observed. Perhaps it would have been good for the study to be carried out on a larger group of patients.
Author Response
Reviewer 2
The chosen topic is particularly interesting. The study is well completed, in which the methodical work can be observed. Perhaps it would have been good for the study to be carried out on a larger group of patients.
Response:
Thank you sincerely for your positive feedback. We conducted this preliminary study with a limited number of participants, and we will continue to expand our research on this topic in the future. This study aimed to explore the relationships between oral microbiota and OSCC-specific pathogens. We kindly ask for your understanding, and once again, the authors would like to express their gratitude for your valuable comments.

Reviewer 3 Report
Comments and Suggestions for Authors
I have gone through the topic. Article is indeed interesting but it appears to be a simplistic description of the preliminary findings. Most probably, these microbes communicate through quorum sensing that plays central role in the regulation of cancers. However, these presumptions and hypothesis need to be experimentally verified. This article does not fit well as a FULL LENGTH RESEARCH ARTICLE. It can be considered as LETTER to EDITOR.
Comments on the Quality of English LanguageModerate language editing is required.
Author Response
Reviewer 3
I have gone through the topic. Article is indeed interesting but it appears to be a simplistic description of the preliminary findings. Most probably, these microbes communicate through quorum sensing that plays central role in the regulation of cancers. However, these presumptions and hypothesis need to be experimentally verified. This article does not fit well as a FULL LENGTH RESEARCH ARTICLE. It can be considered as LETTER to EDITOR. Moderate language editing is required.
Response:
Thank you for your constructive and candid feedback. However, we believe that this study contains detailed analysis results that extend beyond the scope of a letter to the editor. Enrolling both OSCC patients and healthy individuals for simultaneous saliva sampling and analysis is a challenging task. Based on the findings of this preliminary study, we plan to conduct further research with a larger sample size. Additionally, based on your feedback, we have included the statement in the manuscript that "OSCC-specific microbes communicate through quorum sensing, which plays a central role in the regulation of oral malignancy development and progression." We kindly ask for your understanding, and the authors sincerely appreciate your valuable insights. With all due respect, it is not a set rule that studies with a smaller number of enrolled patients must be submitted in the format of a letter. Following your suggestion, we have revised and edited the manuscript for overall language improvements. Thank you again for your thoughtful comments.

Round 2
Reviewer 1 Report
Comments and Suggestions for Authors
The manuscript has been revised properly. Only one minor revision is suggested.
1. Suggestion to the title: Oral microbial changes in Oral Squamous Cell Carcinoma: Focus on Treponema denticola, Lactobacillus casei, and Candida albicans